# Regional occupancy increases for wide-spread species but decreases for narrowly distributed species in metacommunity time series

Wu-Bing Xu [1,2] ✉, Shane A. Blowes [1,2], Viviana Brambilla [3], Cher F. Y. Chow [3], Ada Fontrodona-Eslava[3], Inês S. Martins [3,4], Daniel McGlinn[5], Faye Moyes [3], Alban Sagouis [1,2], Hideyasu Shimadzu [6,7], Roel van Klink [1,2], Anne E. Magurran [3], Nicholas J. Gotelli[8], Brian J. McGill[9], Maria Dornelas [3,4,10] & Jonathan M. Chase [1,2] ✉

While human activities are known to elicit rapid turnover in species composition through time, the properties of the species that increase or decrease their spatial occupancy underlying this turnover are less clear. Here, we used an extensive dataset of 238 metacommunity time series of multiple taxa spread across the globe to evaluate whether species that are more widespread (large-ranged species) differed in how they changed their site occupancy over the 10–90 years the metacommunities were monitored relative to species that are more narrowly distributed (small-ranged species). We found that on average, large-ranged species tended to increase in occupancy through time, whereas small-ranged species tended to decrease. These relationships were stronger in marine than in terrestrial and freshwater realms. However, in terrestrial regions, the directional changes in occupancy were less extreme in protected areas. Our findings provide evidence for systematic decreases in occupancy of small-ranged species, and that habitat protection could mitigate these losses in the face of environmental change.

Humans are accelerating pressures on biodiversity through the Anthropocene due to the confluence of multiple drivers, including habitat loss, climate change, overexploitation, and invasive species[1,2]. A common consequence of this change is that the composition of communities changes through time, where some species can be "winners" and increase through time, while others are "losers" and decrease[3,4]. One prominent hypothesis is that species that have larger ranges and are more widespread tend to be winners during biodiversity change and increase in their abundance and/or occupancy through time, whereas those more narrowly distributed species tend to be losers and decrease in their abundance and/or occupancy through time[5–7]. Reasons for this include the fact that widespread

[1]German Centre for Integrative Biodiversity Research (iDiv) Halle-Jena-Leipzig, Leipzig, Germany. [2]Department of Computer Science, Martin Luther University Halle-Wittenberg, Halle (Saale), Germany. [3]Centre for Biological Diversity, School of Biology, University of St Andrews, St Andrews, Scotland. [4]Leverhulme Centre for Anthropocene Biodiversity, Berrick Saul Second Floor, University of York, York, UK. [5]Department of Biology, College of Charleston, Charleston, SC, USA. [6]Department of Mathematical Sciences, Loughborough University, Leicestershire, UK. [7]Graduate School of Public Health, Teikyo University, Tokyo, Japan. [8]Department of Biology, University of Vermont, Burlington, VT, USA. [9]School of Biology and Ecology and Mitchell Center for Sustainability Solutions, University of Maine, Orono, ME, USA. [10]MARE, Guia Marine Laboratory, Faculty of Sciences, University of Lisbon, Cascais, Portugal. ✉e-mail: wbingxu@gmail.com; jonathan.chase@idiv.de

species tend to have wider niche breadth[8] and more frequent dispersal[9] than more narrowly distributed species, and thus are more likely to persist and/or increase in response to global environmental changes. There is some evidence for this hypothesis, for example, when comparing responses of small- and large-ranged species along land-use gradients[10], and for a few geographically and taxonomically restricted groups through time[5,6,11,12]. However, other studies have failed to find such relationships[13,14], or even find opposite relationships where narrow-ranged species increased through time[15,16]. Thus, the generality of the relationship between range size and the likelihood of a species winning or losing during the Anthropocene remains unclear.

In this study, we evaluated how species' geographic range size is associated with changes in species occupancy within a metacommunity through time (Fig. 1). We define occupancy as the proportion of sites where a species is present in a given year, and thus a species loses occupancy when it occupies a smaller proportion of sites in subsequent time points, and gains occupancy when it occupies a higher proportion of sites in subsequent time points. When anthropogenic pressures favor species that are more widespread and disproportionately disadvantage those that are narrower-ranged, we might expect a positive relationship between range size and temporal changes in species occupancy (Fig. 1b)[5,6,11]. Alternatively, habitat modifications and/or exploitation of widespread species could allow small-ranged species to increase their occupancy[15,16]. Regardless of the direction of change, the removal of anthropogenic pressures, such as by establishing protected areas, would reduce any relationship between range size and occupancy change (Fig. 1b).

To examine how changes in species occupancy are related to species range sizes, we compiled 238 metacommunity time series datasets (i.e., time series of species assemblages from multiple local sites nested within a region) of multiple taxa, including plants, birds, fish, mammals, amphibians and reptiles, and invertebrates, across terrestrial, freshwater and marine realms from three published data sources[17–19], and a fourth data compilation on resurveys that was specifically intended for use in this and related studies (See Methods for the description of the entire dataset). For our analyses, we selected metacommunities that were sampled in at least two-time points with a

minimum of a 10-year time span (10–90 years, median = 16 years), and from at least four local sites within a larger region (4–6308 sites sampled each year, median = 26) (Supplementary Fig. 1). Because some metacommunities were sampled in two years (i.e., the beginning and end of a time series), while others were sampled multiple years throughout the time series, we split those with multiple sampling years into two periods (near the beginning and end of the observation period), calculated the occupancy of a species in each period (the average occupancy from years within the given period), and defined occupancy change as the difference in the occupancy of a species between the late and early periods (see Methods for details). We estimated species' geographic range sizes as the number of 10-km grid-cells across the world occupied by each species using occurrence records from the Global Biodiversity Information Facility[20] (Supplementary Figs. 2 and 3). These estimates were largely robust to the choice of method and size of the grid-cell chosen (Supplementary Fig. 4). For our analyses, we treat species' geographic range size as a static variable because range expansions or contractions for most species should be very small relative to their global ranges during the relatively short monitoring periods of our study (median = 16 years). By contrast, species occupancy within metacommunities can experience substantial changes over a few decades because it is based on species' presence and absence at local sites within relatively small regions[21].

In this study, we assess the overall relationship between species range size and occupancy change across metacommunity time series and compare this relationship across terrestrial, freshwater, and marine realms using hierarchical linear models. For brevity, we will refer to the relationship between range size and occupancy chance as the effect of range size (but note that the analysis is based on observational data). We find an average increase in occupancy through time for large-ranged species and a decrease through time in occupancy for small-ranged species across all metacommunity time series. However, the positive effect of range size differed among realms, with a stronger effect in the marine than in terrestrial and freshwater realms. Furthermore, we find that the relationship between range size and occupancy change is less extreme in metacommunities embedded in areas that receive some degree of protection from human activities relative

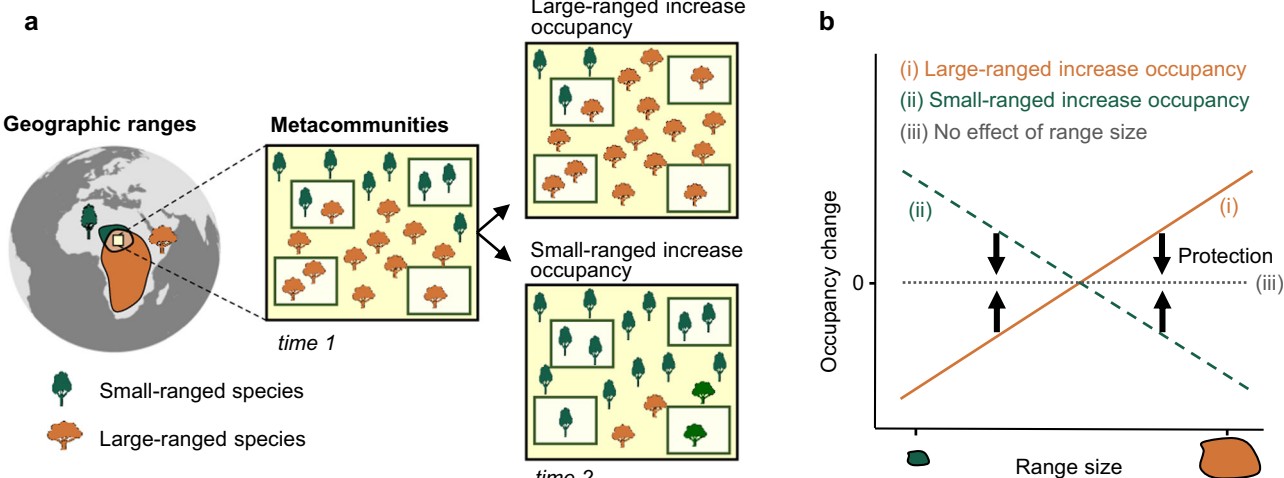

**Fig. 1 | Hypothetical relationships between species' geographical range sizes and temporal changes of occupancy. a** By comparing metacommunities surveyed in two time periods, we illustrate two contrasting scenarios about how two species with different range sizes might change their proportional occupancy through time: (i) large-ranged species increase occupancy but small-ranged species decrease occupancy; (ii) small-ranged species increase occupancy but large-ranged species decrease occupancy. Other scenarios, such as both small- and large-ranged species decreasing or increasing occupancy with differing magnitudes, are not

shown for clarity. **b** Changes of species occupancy through time as a function of their range sizes from the scenarios illustrated in **a**. The effect of range size on occupancy change is expected to be reduced or removed (iii) in protected areas if the protection works (black arrows in **b**). For simplicity, the average changes of occupancy are shown as zero when range size has no effect, although it is possible that habitat protection can increase the occupancy of both small and large-ranged species.

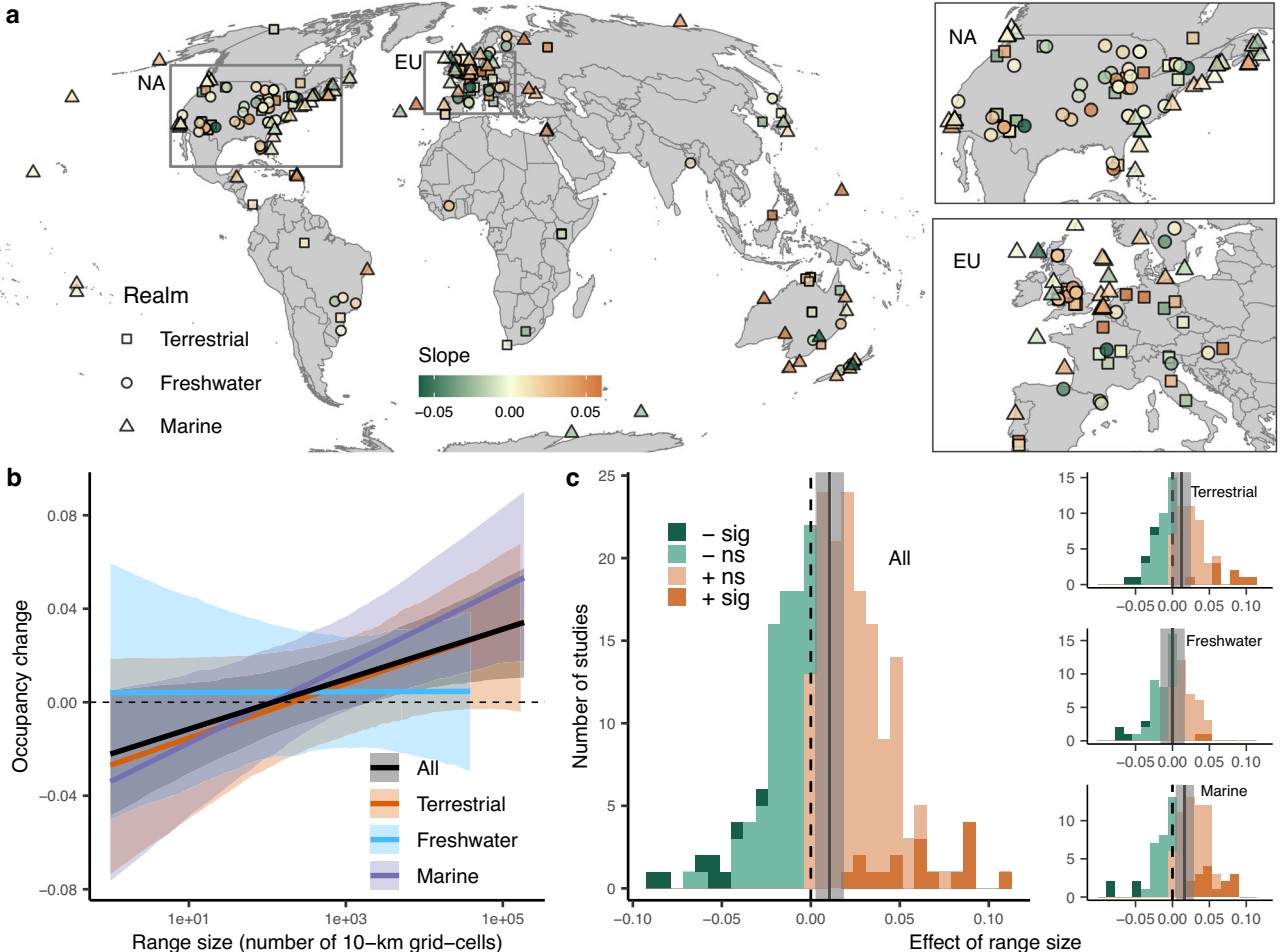

**Fig. 2 | Small-ranged species lose, but large-ranged species gain occupancy over time. a** Global map showing the distribution of effects of range size (slope) on occupancy change of 238 studies in terrestrial (*n* = 81, squares), freshwater (*n* = 68, circles) and marine (*n* = 89, triangles) realms. Inserts show detail for North America (NA) and Europe (EU). For clear visualization, the slopes smaller than −0.06 or greater than 0.06 were rounded to −0.06 or 0.06. **b** Changes in species occupancy as a function of species' range sizes. Occupancy changes are the difference in occupancy between the late and early periods, shown as the square root- transformed for the absolute magnitude. The black line and shading show the overall positive relationship and 95% credible interval; colored lines and shading indicate

the relationship for terrestrial (orange), freshwater (blue) and marine (purple) realms, estimated with a separate model. **c** Frequency distribution of study-level slope estimates for all studies combined and different realms. Solid lines and shadings show the overall slope estimate and 95% credible interval. The dashed line shows the zero slope (no effect of range size). Bars are color-coded as dark orange (positive, significant), light orange (positive, non-significant), light green (negative, non-significant), and dark green (negative, significant) based on the sign of each study-level slope estimate and whether its 95% credible interval overlaps zero. See Supplementary Tables 1 and 2 for model summaries and sample sizes.

to metacommunities in unprotected areas within the terrestrial realm, suggesting that habitat protection could mitigate systematic biodiversity changes in the Anthropocene.

## Results and discussion
### Relationships between species range size and occupancy change
We found strong evidence for a positive relationship between species range sizes and occupancy changes through time across the metacommunities (slope = 0.011, 95% credible interval (CI): [0.003, 0.019]); Fig. 2 and Supplementary Table 1). That is, large-ranged species tended to gain in occupancy, while small-ranged species tended to decline in occupancy through time (Fig. 2b). This result is consistent with several smaller-scale regional analyses[5–7,11,12], and may reflect the capacity of large-ranged species to tolerate changing environmental conditions as a result of broader environmental tolerances (e.g., generalist strategies), and/or greater dispersal capacities[8,9].

While we found an overall positive effect of range size on changes in occupancy, there was considerable variation in the slope estimates of individual studies even among geographically adjacent regions

(Fig. 2a, c). Specifically, of the 238 studies we analyzed, 155 studies had positive slopes, but only 17 of them differed from zero based on their 95% credible intervals. On the other hand, 83 studies had negative slopes, but only 8 of those differed from zero based on their 95% credible intervals. This suggests that despite the overall positive effect of range size on occupancy change, different metacommunities conform to each of the hypotheses illustrated in Fig. 1 to varying degrees. However, it is also clear that ecological processes other than range size influence changes in species occupancy over time. For example, intrinsic population fluctuations are prevalent in many natural assemblages, which can drive species turnover without external environmental changes[22]. In addition, biological traits other than range size can influence how species change in their occupancy through time. For example, warm-adapted species are favored relative to cold-adapted species under climate warming[23]. Likewise, habitat modification of forested ecosystems favors non-forest and disturbance-adapted species relative to forest-dwelling and disturbance-intolerant species[24]. As a result, even though our synthesis based on hundreds of heterogeneous studies shows an overall positive

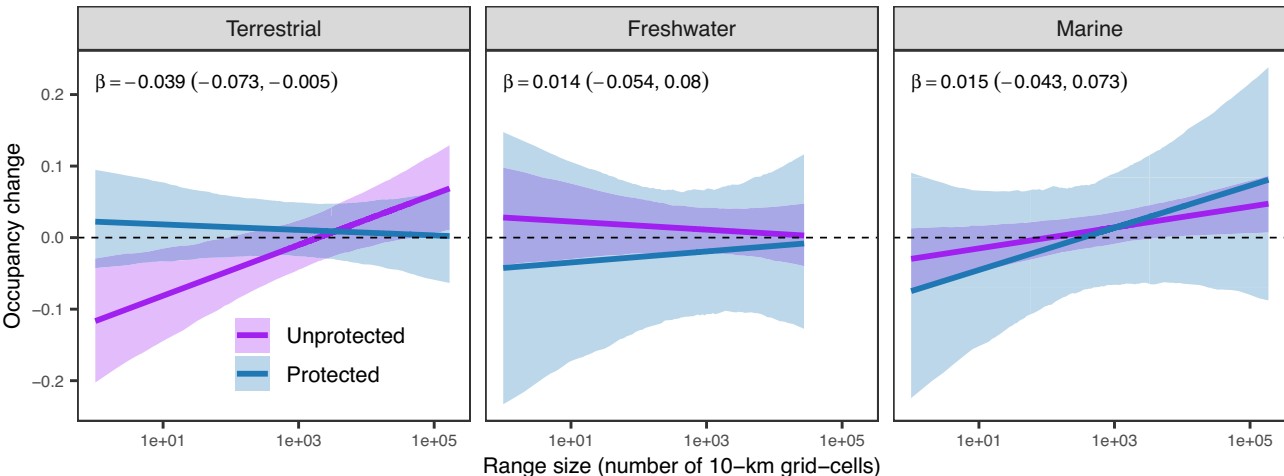

**Fig. 3 | Effect of habitat protection on the relationship between range size and occupancy change for terrestrial, freshwater, and marine realms.** The coefficient (β) of the interaction between range size and proportion of sites that occur within protected areas for each realm and its 95% credible interval are shown at the top. The purple solid lines show the predicted relationship when no sites within a metacommunity are protected, while the blue solid lines show the predicted relationship when all sites within a metacommunity are protected; the shading shows the 95% credible intervals. The model summary and sample size are in Supplementary Table 3.

relationship between range size and occupancy change, the large variation around this trend is not unexpected and requires deeper exploration.

One potential limitation of our study could emerge if most distribution records in GBIF we used to estimate range sizes were contributed by the same assemblage data we used to estimate occupancy change (e.g., some time series from the BioTIME dataset[17] were sourced and/or similarly contributed to GBIF). To evaluate whether this sort of non-independence among the datasets might have influenced our results, we compared the number of occurrences in the two data sources (Supplementary Fig. 5), and repeated the analyses using only species that have at least five times more records in GBIF than records in our metacommunity assemblage data (13,876 of 16,651 species, 83.3%). We found a similar effect of range size using this subset of data (Supplementary Fig. 6a). Additionally, we found qualitatively consistent results when range size was measured as the number of 50-km, or 100-km grid-cells, or the extent of occurrences (Supplementary Fig. 7; see Methods for details on different measures of range size). These sensitivity analyses suggest that the positive effect of range size on occupancy change was robust to possible uncertainty in estimating range size. Furthermore, our results were robust to the approach used to filter sites within the metacommunity (see *Data standardization* in Methods; Supplementary Fig. 6b).

### Variation in range-size occupancy-change relationships across realms

While our results showed that the overall effect of range size on occupancy change was positive, there were differences across realms: marine systems had the steepest slopes (slope = 0.017, 95% CI: [0.004, 0.029]), terrestrial systems had moderate slopes (differed from zero with only 90% certainty; slope = 0.012, 90% CI: [0.001, 0.023], 95% CI: [−0.002, 0.025]), and freshwater systems showed no effects (slope = 0.000, 95% CI: [−0.016, 0.017]) (Fig. 2b, c and Supplementary Table 2). The steep slopes in marine systems are consistent with the observation that species compositional turnover is also higher in marine systems[25]. One potential reason could be that species sensitivities to environmental changes (e.g. climate warming) are higher in the marine realm[26] and there are fewer dispersal barriers. Freshwater systems, on the other hand, showed no relationships between range size and occupancy shifts, possibly because range sizes of freshwater species are mainly determined by the hydrological connectivity of drainage

basins[27], which could weaken their biological capacity to respond to environmental changes. Freshwater results might also differ because habitat conditions in some areas have benefited from restoration and mitigation efforts[28], allowing some taxa to increase[29]. Despite the considerable variation, we found that the differences among realms were largely consistent among taxonomic groups and geographic regions (Supplementary Figs. 8 and 9). For the few instances with anomalous results, they usually had few studies and thus low confidence. For example, our analyses suggested an opposite, negative, association in South American freshwater, however, only four datasets were sampled in that region.

### Effectiveness of protected areas

To mitigate the potential influence of anthropogenic pressures on biodiversity, humanity has designated ~17% of terrestrial areas and inland waters and ~8% of coastal and marine areas[30] as 'protected' in some way. While there is some evidence that protected areas effectively conserve biodiversity[31,32], these designations do not protect against all human pressures, which could reduce their effectiveness[33]. To determine how the establishment of protected areas might mitigate the influence of range size on occupancy change, we quantified the proportion of surveyed sites within a given region (i.e. metacommunity) that were assigned some sort of protected status prior to the sampling in the late period (Supplementary Fig. 10). We found a negative interaction between range size and the proportion of sites with some protection status on occupancy change in the terrestrial, but not freshwater or marine, realms (Fig. 3 and Supplementary Table 3). That is, in terrestrial systems, the relationship between range size and occupancy change tends to be less extreme when regions are more protected compared to non-protected regions (Fig. 3). This suggests that habitat protection might be successful in stemming some aspects of biodiversity change, in particular, by minimizing the increase in occupancy of large-ranged species and the decrease in occupancy of small-ranged species in terrestrial ecosystems. This is consistent with observations that there are often more threatened and endemic species[32] and fewer invasive species[34] in protected areas, which may prevent some threatened and endemic species with small ranges from extirpation and invasive species with large ranges from expanding in protected areas.

Our results on the influence of protected areas were consistent when we used all protected areas, including those that were

established more recently (Supplementary Fig. 11), and when the degree to which a region is protected was measured as the proportion of the spatial extent of a given metacommunity covered by protected areas (Supplementary Fig. S12). Because the effect of protection status may be mediated by other study characteristics (Supplementary Fig. S13), we fit a model considering protection status together with six other variables, including latitude, regional species richness, spatial extent of sampled sites, number of samples, duration and start year of sampling. We found a consistent effect of protection to mitigate the relationship between range size and occupancy change in the terrestrial realm, whereas all other variables except regional species richness and start year of sampling had no significant effects (Supplementary Table 4). The difference in the effect of protection among realms may result because many marine and freshwater protected areas have not reached their full conservation potential, for example, due to difficulties in enforcing protection and/or emigration of animals outside protection boundaries in highly interconnected habitats[35,36]. In addition, compared to terrestrial regions, only a few regions from the marine and freshwater realms were situated within protected areas (Supplementary Fig. 10), which could reduce our ability to detect any influence of protection in marine and freshwater realms.

## Caveats and implications

Overall, our results support the hypothesis that small- and large-ranged species differ in how they are responding to the last several decades of the Anthropocene. These results generalize previous smaller-scale studies[5–7,11,12] to suggest this is a common phenomenon. With our current analyses, however, we cannot directly address the drivers of this result, nor can we specifically determine potential mechanisms behind this overall result for a multitude of taxa, geographic regions, and scales on which the individual studies took place. Nevertheless, we suggest that it is likely that the trends we observed emerged because ongoing environmental change have small, or positive, effects on widespread species with broader habitat breadth, while small-ranged species with narrower habitat breadth may be disfavored under anthropogenic pressures[8,37,38]. Concurrent with differences in habitat breadth, larger-ranged species also tend to have stronger dispersal abilities, at least in some cases[9,39]. Both broader habitat breadth and higher dispersal rates may be associated with the positive relationships we observed for larger-ranged species increasing occupancy over time. For example, species with larger habitat breadth, dispersal rates and native range sizes are more likely to become naturalized and invasive outside of their native range, leading to increases in occupancy through time[40,41]. It is also important to note that range size, which we have used as a predictor of species occupancy change, is an emergent property of species that results from underlying mechanisms associated with species traits and biogeographic constraints[8,42–44]. Past human pressures that occurred well before the temporal trends analyzed here are also associated with differences in species range sizes[45] and could have influenced our results. To more fully understand the mechanisms underlying our findings, it would be important to clarify how underlying biological traits influence range size, which in turn influences the likelihood of gaining or losing occupancy through time. For example, in addition to habitat breadth and dispersal ability, several other traits seem to be associated with range size at least in some taxa and systems, including body size, fecundity, longevity, habitat preference, and nutrient demands, among others[6,42,46–49]. However, a synthetic analysis across multiple taxa and realms, such as what we have done here, would require comparability among those traits in order to detect general patterns, which is challenging, if not impossible. Nevertheless, some traits, such as body size, incorporate numerous underlying traits (e.g., life history) and can be compared across taxa, providing a potentially important next step for analyses of species occupancy change through time.

Although our focus here has been on changes in occupancy through time at the species level, our results have implications for understanding the directional compositional turnover of communities. Globally, the turnover of species composition in communities through time tends to be occurring more rapidly than expected due to random chance[25,50]. What has been less clear is whether there are species characteristics that lend advantages or disadvantages in response to environmental change through time, which might ultimately lead to scale-dependent biodiversity changes[51] and/or biotic homogenization[3,52]. Our results show a general trend that smaller-ranged species are decreasing in occupancy through time, and that these tend to be replaced by larger-ranged species that are increasing in occupancy through time. Finally, because we found that directional changes in species distributions are weakened in regions that have higher levels of protection, at least in the terrestrial realm, we confirm that habitat protection can provide important mitigation against systematic biodiversity changes in the Anthropocene.

## Methods

### Temporal metacommunity assemblage data

To analyze temporal changes of species occupancy within a metacommunity (i.e., multiple localities within a region), we searched for datasets that had survey data of an assemblage sampled using a similar methodology from at least four local sites each year, spanning at least 10 years between the first and last sampling date. We extracted data from four open-access databases of compiled assemblage time series from sites across the world, including BioTIME[17], RivFishTIME[18], InsectChange[19], and a previously unpublished database (hereafter Metacommunity Resurveys[53]). Metacommunity Resurveys was specifically compiled for synthesizing patterns of temporal change in metacommunities (for this and related studies), with most of its studies designed to 'resurvey' sites that had been surveyed 10+ years previously using similar methods. While RivFishTIME focusses specifically on fish in streams and rivers, and InsectChange focusses specifically on insects, the BioTIME and Metacommunity Resurveys compilations contain data on various taxa (e.g., plants, mammals, birds, invertebrates) from multiple realms (terrestrial, freshwater and marine).

In all, we compiled a total of 238 studies (see Supplementary Table 5 for the list of studies and citations of original studies), including 81 terrestrial, 68 freshwater and 89 marine studies. One hundred of these studies came from BioTIME[17], 34 came from RivFishTIME[18], 23 came from InsectChange[19] and the remaining 81 studies came from the Metacommunity Resurveys compilation[53]. Here, "study" refers to data on surveys of assemblages over time at multiple-sites within a region (i.e., a metacommunity time series; usually collected by the same team of investigators and/or using the same methodology). For each study, we ensured that the sampling effort was largely equivalent across time, or could be standardized to be so. If two or more regions or taxonomic groups were present in the same dataset, they were treated as separate studies. The datasets including multiple studies were usually compilations of original surveys from different regions. There were only seven regions that contained surveys on two or three taxonomic groups, contributing to a total of 15 studies. Our selected studies had a median of 10 sampling time-points (2–75 points), and a median of 26 sites (4 – 6308 sites) at each time point. The data spanned from 1927 to 2021, with a median start year of 1994, and a median duration of 16 years, ranging from 10 to 90 years (Supplementary Fig. 1).

We examined species dynamics through time within each metacommunity. By comparing species presence and absence and occupancy changes between late and early periods (see 'Calculating occupancy'), species were classified into five groups: lost (present in the early period, but absent in the late period), gained (present in the late period, but absent in the early period), persisted with increased

occupancy (present in both periods, but higher occupancy in the late period), persisted with decreased occupancy (present in both periods, but higher occupancy in the early period), persisted with stable occupancy (present in both periods with no occupancy changes). Across our dataset, we found a median of 16.7% gained species (percentage of all species found in a given metacommunity; the same below), 12.0% lost species, 26.5% persisted species with increased occupancy, and 24.3% of persisted species with decreased occupancy (Supplementary Fig. 14). These strong species dynamics allow us to detect the effects of species' range size on occupancy change through time.

### Harmonizing taxon names

To account for changes in taxonomy across surveys, we determined accepted species names for each species based on the taxonomy from the GBIF backbone[54]. We used the GBIF backbone because it provides a synthetic classification for all taxonomic groups, and the accepted species names were also used to extract occurrences from GBIF to estimate species' range size. Because not all taxonomic names in the original datasets were identified to species, 19,110 of 25,607 (74.6%) names were standardized as a species in the GBIF backbone. Of these standardized species, 99% (18,914) species have distribution occurrences in GBIF. We selected studies that had at least 10 species with range size estimates for our analyses. In the selected studies, most of the sampled species have range size estimates, with the median of the proportion of species within studies that had range size estimates as 91% (Supplementary Fig. 15a). Further, there was no correlation between the proportion of species within studies that had range size estimates and the study-level effect of range size on occupancy change (Supplementary Fig. 15b), suggesting missing some species without range size estimates had little influence on our results.

### Data standardization

Because the number and spatial configuration of sites can influence estimates of species occupancy, and most studies had different numbers and locations of sites across years, we developed an approach to maximize the number of sites in an approximately constant spatial configuration sampled at least two years for each study. We first subdivided the sampled extent into grid-cells with a resolution defined as 1/5 the mean of the longitudinal and latitudinal spans of each study. The selected resolution was a trade-off between coarse resolutions which would reduce abilities to choose sites in similar spatial configurations across years and finer resolutions which would result in the loss of many sites that cannot be matched through years (see next paragraph for a sensitivity analysis of this approach). We kept all cells that contained surveys across all years, and cells with more than half the average number of sites across cells. We then calculated the number of sites that co-occurred in the same cells between all possible year-pairs with an interval longer than 10 years. We determined which two years had more co-occurring sites than the 90% of the maximum co-occurring site number and spanned a longer duration (to maximize both spatial and temporal extent in a given study). We selected the cells that contained surveys in both determined years. Other years were compared to the two determined years, and we kept the years that had sites in more than 90% of selected cells (weighted by the relative number of sites in each cell). This resulted in a dataset with sites in similar spatial configurations across years. However, the number of sites in the filtered dataset may differ across years. We thus used sample-based rarefaction to account for the variation in the number of sites (see 'Calculating occupancy'). Prior to this, we excluded years with fewer than half the mean number of sampled sites, providing the remaining years surveyed spanned at least 10 years.

To test the sensitivity of our results to this approach of site-matching, we further filtered the above dataset by keeping only sites that have the exact geographic coordinates across years. Similar to the

primary approach, we compared locations of sites between all possible year-pairs with an interval longer than 10 years, and determined which two years had more sites in the same locations than the 90% of the maximum number of same-location sites and spanned a longer duration. We then selected the shared sampled sites in the two determined years and kept the years that sampled all these sites. Compared to the 'grid-based' filtered dataset, 32 studies were lost in the resultant dataset, of which 31 were from the marine realm, leaving a total of 206 studies; the number of samples (samplings at a location in a year) was reduced from 727,356 to 41,774 for all studies, from 552,943 to 8685 for marine studies, from 90,402 to 10,487 for freshwater studies, and from 84,011 to 22,603 for terrestrial studies.

We also standardized the sampling effort so that it was as consistent as possible across all sites and years. Some studies collected different numbers of samples (e.g. transects) across sites and years, in which we standardized using sample-based rarefaction[55] to randomly select an equal number of samples, which were then combined to provide one sample per year for each site. Some studies used different sampling methodologies for different sampling events (e.g. seining and electrofishing for collecting freshwater fish). For these, we identified the methodology that was used in the greatest number of sites within a given region, and standardized sampling efforts only using the data collected with that methodology.

### Calculating occupancy change

We first calculated each species' occupancy within each metacommunity in each year. Occupancy was defined as the number of sites where a species was present in a given year divided by the total number of sites surveyed in that year. While many of the studies provided data that were collected periodically throughout the time series[17–19], a substantial proportion of the studies in our data compilations were based on resurveys of historical data where surveys only took place during only two time points[56,57] (e.g., historical versus recent periods). Because of this, and to provide more straightforward analyses and interpretations of all datasets, we focused our analyses on changes in occupancy between two periods–near the beginning and end of the time series. For studies that had only two-time points, we used data from the first and last years to calculate occupancy in the early and late periods. For studies that had data from more than two-time points, we used the middle year of a given time series and defined the years before the middle year as the early period and the years after the middle year as the late period. Because many time series did not collect data regularly (e.g. every year, every two years), we standardized sampling effort (number of years) before and after the mid-point by taking the minimum number of years sampled in either the early or late periods and selected that number of years from the other period. For the period that had more time points sampled, we used the earliest years needed for the early period or the latest years needed for the late period to maximize the duration between observations. Of the 238 studies, 177 of them had multiple years in both early and late periods. For these studies, the median of the number of years per period was 5, with a range of 2 to 35 years. We defined the occupancy of each species for a given period that had samples from multiple years as the average occupancy of that species across those years. For a site that was observed for several years during a given period, it contributed multiple samples in the calculation of average occupancy. Here, a sample was defined as an observation event of a local site within a year. That is, the occupancy of a species in a given period was calculated as the number of samples where a species was present across the years in that period divided by the total number of samples in the same period. We then calculated the change in each species' occupancy as the differences in its occupancy between the late and early periods. Occupancy changes varied from −1 to 1, with a value of −1 indicating a species was present in all samples in the early period, but absent from all samples in the late period, and a value of 1 indicating a

species was present in all samples in the late period, but absent from all samples in the early period.

In the calculation of occupancy, we accounted for variation in the number of sites sampled across years by calculating the minimum number of sites in a year and randomly selecting this number of sites from each time point. We repeated this rarefaction process 200 times to test whether our results were robust to the random samples selected. In each iteration, we calculated species occupancy and tested the relationship between range size and occupancy change. Because different species sets can be sampled across rarefaction iterations, we did not calculate the mean species occupancy across iterations. We reported results based on one iteration in the main text and showed the frequency distribution of overall slope estimates of the relationship between range size and occupancy change across 200 iterations in Supplementary Fig. 16.

## Estimating range size

To estimate each species' range size, we compiled occurrences from the Global Biodiversity Information Facility in December 2022[20]. We cleaned occurrence records by excluding: (a) records without coordinates; (b) records based on fossils, material samples, and living collections; (c) records with reported uncertainty in coordinates larger than 100 km; (d) records located at country centroids, capitals, biodiversity institutions, localities with equal latitude and longitude, and zero coordinates; (f) duplicated records of species within grid-cells of $0.01°$. We further excluded records in the sea for terrestrial and freshwater species, and records on land for marine species using the 10-km buffered worldwide land and ocean maps. The final dataset includes 189,370,752 occurrence records for 18,715 species. We used the R package 'rgbif'[58] and 'CoordinateCleaner'[59] to download and clean occurrences.

Based on GBIF occurrences, we estimated species' range size in two ways: area of occupancy (AOO) and extent of occurrences (EOO)[60] (Supplementary Fig. 2). We estimated AOO as the number of grid-cells that were occupied by a species. This approach tends to underestimate ranges due to incomplete sampling, but avoids overestimating ranges due to fragmented and noncontinuous distributions. Furthermore, AOO is less sensitive to outliers than EOO. We estimated AOO using three grid-cell resolutions: 10 km × 10 km, 50 km × 50 km, and 100 km × 100 km. These and similar resolutions have often been used when estimating geographic range size in similar studies[10,11]. Following previous studies[61,62], we estimated EOO as the area of alpha hulls for species with more than three occurrences. The alpha hull is a generalization of the convex hull[63] and allows the constructed geometric shape to be several discrete polygons dependent on the value of the parameter alpha. We used the alpha value of six to construct alpha hulls, as recommended by ref. [61], using the R package 'alphahull'[63]. For species with fewer than three records, the summed area of 10-km buffers around each point was used to estimate their EOO, as performed in ref. [64].

For each species, we calculated one value for each measure of range size, which was assumed to have no or only very small changes during the relatively short monitoring periods of our study (median = 16 years), and thus regarded as a static measure. We acknowledge that species' geographic range size can change over time, but range expansions or contractions for most species should be very small relative to their global ranges at the time scale of assemblage monitoring periods in this study. By contrast, species-proportional occupancy within a region can experience substantial changes during a few decades.

Although there is uncertainty in the estimates of geographical range size due to sampling bias in GBIF, this is unlikely to influence our results because we only directly compared range sizes of species within studies and our hierarchically structured models accounted for differences across studies. That is, we only require estimates of relative differences in range sizes of species within the same taxonomic group and region from individual studies. In addition, most of the studies in our dataset came from Europe ($n = 45$) and North America ($n = 65$), as well as the Atlantic ($n = 50$) and Pacific ($n = 33$) Oceans (usually located on the coast and near offshore), where GBIF occurrences have relatively good coverages (https://www.gbif.org). Moreover, there were strong correlations (Pearson's $r > 0.87$) among different estimates of range size calculated in this study (Supplementary Fig. 4). We showed results based on AOO estimated at the resolution of 10-km in the main text and performed sensitivity analyses using other estimates of range size. In this study, we did not use IUCN range maps to calculate range size because they were not available for all of our study species (e.g. most invertebrates, fishes and plants) and previous studies have shown that range size estimates based on GBIF occurrences and IUCN range maps were strongly positively correlated[10,11].

## Protected status

Because habitat protection is an important strategy for minimizing potential biodiversity change, we tested whether the relationship between range size and occupancy change varied with the degree to which a given study region (i.e. metacommunity comprising multiple local sites) is protected. We obtained data on protected area boundaries and year of inscription from the 2022 World Database on Protected Areas (WDPA)[65]. Following recommendations in the WDPA website (https://www.protectedplanet.net/en/resources/calculating-protected-area-coverage), we extracted those protected areas that have a status of "designated", "inscribed", or "established", and were not designated as UNESCO Man and Biosphere Reserves. We used protected areas with detailed geographic boundaries (polygons) and those represented in points. For the protected areas represented by a single point, we generated a buffered area around each point with an area equal to the recorded area. There were many protected areas that were established recently, particularly marine protected areas, whose area coverage increased from <1% in 2005 to -7.6% by 2021[65]. We thus only included protected areas that were established prior to the sampling in the late period of each metacommunity when evaluating whether a site was given protected status for our main analysis. However, we also performed sensitivity analyses using all protected areas, including those that were established more recently, because protected areas are usually established in high-quality ecosystems and those recently established protected areas were likely already of high quality before given official protection status.

For each study, we calculated the number of local sites that fall within protected areas and then divided that by the total number of sites within a given region (Supplementary Fig. 10). The proportion of sites that were given some sort of protected status was used to represent the degree to which a region is protected. Because species can disperse across continuous space, species occupancy within a given metacommunity is possibly affected by the protection status of the whole spatial extent of a metacommunity. To represent the spatial extent of a given metacommunity, we constructed a convex hull comprising all local sites in each metacommunity. We cropped the convex hull to keep only areas on the land for terrestrial and freshwater metacommunities and only areas in the sea for marine metacommunities. We overlaid polygons of protected areas over each cropped convex hull of the metacommunity to calculate the proportion of the spatial extent of the metacommunity covered by protected areas (Supplementary Fig. 10). We used the proportion of sites in a metacommunity that fall within protected areas in the main analyses and the proportion of the spatial extent covered by protected areas in a sensitivity analysis.

There were 43 studies in our data where we only had a single, central coordinate for the region, but no coordinates for individual local sites. For these studies, we determined the protected status of these regions based on regional descriptions in original papers. If a

region was protected overall, all sites in the region were regarded as protected. If a region was influenced by strong human activities (e.g. farming), no sites in the region were regarded as protected. For regions whose protected status cannot be determined based on regional descriptions, we determined their protected status based on whether their regional central locations were located in protected areas if the regional extents were less than 10 km²; but for regions with large extents (>10 km²), we left the protected status as 'unknown' (20 studies), and these studies were excluded from the analysis of the effectiveness of protected areas.

## Statistical analyses

We analyzed the relationship between species' geographic range sizes and temporal changes in species occupancy using hierarchical linear models. Because most values of occupancy change were distributed around zero, we fit and compared models regressing occupancy change as a function of range size using three error distributions (Gaussian, asymmetric Laplacian and Student's t) in preliminary analyses. However, all these models do not well describe the empirical distribution of occupancy changes based on posterior predictive checks using the 'pp_check' function in the R package 'brms'[66] (Supplementary Fig. 17). To decrease the kurtosis of occupancy changes, we first square root-transformed the absolute value of occupancy change, and then multiplied that by the sign of the change (termed as 'sign*square root-transformation') (see e.g., Jandt et al.[67] for a similar usage). These transformed occupancy changes still had values ranging from −1 to 1. Range size was log$_{10}$-transformed and centered by subtracting the mean (log) range size before fitting models, and back-transformed for presentation in figures. We estimated the overall relationship between range size and occupancy change, while allowing the intercept and slope of the relationship to vary for each study. That is, the intercept and slope of the relationship were estimated as fixed effects and also random effects for each study. We also allowed the variance of occupancy change to vary across studies. The variance of occupancy change was expected to decrease with the number of samples that were used to calculate occupancy in each period. For example, if a species in a metacommunity with four samples in each period went randomly extinct in one sample, its occupancy would decrease by 0.25; while a species would decrease occupancy by only 0.01 if it randomly went extinct from one sample in a metacommunity with 100 samples in each period. We thus modeled the logarithm of standard deviation of occupancy change as a function of the log-transformed number of samples that were used to calculate occupancy. The model had the form:

$$\Delta p_{ij} \sim \text{Normal}(\mu_{ij}, \sigma_i)$$
$$\mu_{ij} = (\beta_0 + u_{0i}) + (\beta_1 + u_{1i})\text{range}_j,$$
$$\log(\sigma_i) = \eta_0 + \eta_1 \log(\text{nsamp}_i)$$

Where $\Delta p_{ij}$ is the sign*square root-transformed occupancy change for the $j$th species in the $i$th study; range$_j$ is the log$_{10}$-transformed and centered range size for the $j$th species; nsamp$_i$ is the total number of samples in each period in the $i$th study; $\beta_0$ and $\beta_1$ are the global intercept and slope (fixed effects) that estimate mean occupancy change ($\mu_{ij}$); $u_{0i}$ and $u_{1i}$ are the departures of study-level intercepts and slopes from the $\beta_0$ and $\beta_1$, respectively; $\eta_0$ and $\eta_1$ are the intercept and slope to fit the logarithm of standard deviation of occupancy change ($\sigma_i$).

To examine the variation in the effect of species range size on occupancy changes across studies, we first extracted the slope estimates of the individual studies from the above hierarchical linear model. We used the 95% credible interval of each study-level slope estimate to determine which studies had positive or negative slopes that were different from zero.

Second, we examined variation in range-size occupancy-change relationships associated with realms and geographic regions where studies were located, as well as the taxonomic groups considered. We classified realms into terrestrial, freshwater and marine. Geographic regions for terrestrial and freshwater studies were classified according to the continents where studies took place: Africa, Asia, Australia, Europe, North America, and South America. Marine studies were classified according to the oceans where the studies took place: Arctic Ocean, Atlantic Ocean, Indian Ocean, Pacific Ocean, and Southern Ocean. As there were few studies located in the Arctic Ocean ($n = 2$), Indian Ocean ($n = 3$), and Southern Ocean ($n = 1$), they were combined into one group for analyses. Taxa were lumped into the following groups: amphibians and reptiles, birds, fish, invertebrates, mammals, and plants. Seven marine studies classified as 'multiple taxa' and 11 marine studies classified as 'benthos' in BioTIME included multiple taxonomic groups but had one group that usually dominated. To simplify interpretation, we reclassified the taxa of these studies into the taxonomic group that included most species within each study (see Supplementary Table 5 for these studies). As we were primarily interested in comparing the effects of range size across realms, we fit a model with a two-way interaction between range size and realm. To evaluate how the effects of range size varied across geographic regions and taxa in different realms, we fit models with three-way interactions among range size, realm and region (or taxon) in preliminary analyses. However, these models did not converge, probably because not all regions or all taxa had data in each of the three realms. We thus combined realms and regions (realm-region), and realms and taxa (realm-taxon), and fit models with a two-way interaction between range size and realm-region (or realm-taxon). All models with interactions included an intercept and slope (fixed effects) for each categorical group.

Third, we evaluated how the establishment of protected areas can affect the relationship between range size and occupancy change in different realms. We fit a model with an interaction between range size and the proportion of sites within a given region that were given some sort of protected status for each realm (i.e. three-way interaction: range size*protection*realm). When the main effect of range size on occupancy change was positive, a negative interaction between range size and protection level would suggest that the relationship between range size and occupancy change tends to be weaker in regions that received higher protection. In the main results (Fig. 3), we reported the results from a model using the proportion of sites in those protected areas that were established prior to the sampling in the late period. However, for sensitivity analyses, we also provided results in the supplement using the proportion of sites in protected areas established at any time points (Supplementary Fig. 11). We also provided a sensitivity analysis using the proportion of the spatial extent of a metacommunity covered by protected areas instead of the proportion of sites that fall within protected areas (Supplementary Fig. 12). To evaluate whether the interaction between range size and protection level was influenced by other covariates, we fit another model including additional interactions between range size and each of six study characteristics for each realm. These six study characteristics included the absolute central latitude of study sites, the total species richness within a given region, the extent of sampled sites (the area of the convex hull incorporating all sites or the bounding box from original studies when local coordinates were not available), the number of samples used to calculate occupancy in each period, the duration and start year of sampling (Supplementary Fig. 1).

## Model fitting

We employed Bayesian inference to fit all models using the R package 'brms'[66]. Models were run using 4 chains, each with 8000 iterations, with a warm-up of 4000. We used the default non-informative brms priors for all parameters. We assessed convergence by using Rhat

values (Gelman-Rubin diagnostic) and visually examining Markov chains. All analyses were run in R v.4.0.2[68].

## Sensitivity analyses

We evaluated the sensitivity of our results to how we matched sites through years and how we estimated range size. To include as much data as possible in analyses, we used the 'grid-based' filtered dataset in main analyses. To test the sensitivity of our results to the data filtering approach, we repeated analyses using the subset that contained sites in the exact same locations across years. Due to uncertainty in estimates of range sizes, we calculated four values of range size for each species using different measures and spatial resolutions. To test whether our results were sensitive to these estimates, we repeated analyses using all other estimates of range size besides the AOO based on the resolution of 10-km used in the main analyses, and reported the overall effect of range size and study-level slope estimates (Supplementary Fig. 7). Because at least some of the assemblage-level datasets used to estimate occupancy change are also present in GBIF (e.g., some time series from BioTIME were sourced and/or similarly contributed to GBIF), there could be some concerns about circularity in our comparisons of occupancy change and range size. This was not likely the case, however, because we found that when we compared the number of occurrences (number of occupied 0.01° grid cells) for each species using these two datasets, most species (17,720 of 18,715, 94.9%) had considerably more occurrences in GBIF than the assemblage-level dataset (Supplementary Fig. 5). Nevertheless, some species had more records in the assemblage dataset, whose range sizes were probably underestimated. We thus tested the sensitivity of our results including only the species that had at least five times more occurrences in GBIF than in the assemblage-level dataset in statistical analyses.

## Reporting summary

Further information on research design is available in the Nature Portfolio Reporting Summary linked to this article.

## Data availability

All of the data used in this analysis are open access and available on GitHub (https://github.com/Wubing-Xu/Range_size_winners_losers), and are mirrored on Zenodo (https://doi.org/10.5281/zenodo.7675355)[69]. Additionally, the original sources of data used in this study are publicly available. The BioTIME data can be accessed on Zenodo (https://doi.org/10.5281/zenodo.2602708)[17] or through the BioTIME website (https://biotime.st-andrews.ac.uk/); the RivFishTIME data can be accessed through the iDiv Biodiversity Portal (https://doi.org/10.25829/idiv.1873-10-4000)[18]; the InsectChange data can be accessed on KNB (https://doi.org/10.5063/F11V5C9V) or through the data paper (http://onlinelibrary.wiley.com/doi/10.1002/ecy.3354/suppinfo)[19]; the 'Metacommunity Resurveys' data can be accessed through the iDiv Biodiversity Portal (https://doi.org/10.25829/idiv.3503-jevu6s)[53]; the species occurrences are available on Global Biodiversity Information Facility (https://doi.org/10.15468/dl.6vdkbn)[20]; the protected area data are available on World Database on Protected Areas (accessed October 2022; https://www.protectedplanet.net/en/thematic-areas/wdpa?tab=WDP)[65].

## Code availability

The R code used for standardizing the data and doing the analyses presented here are available on GitHub (https://github.com/Wubing-Xu/Range_size_winners_losers), and are mirrored on Zenodo (https://doi.org/10.5281/zenodo.7675355)[69].

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

## Acknowledgements

We thank all of the authors who contributed their data or gave us permission to include it in this study. Conversations with members of the extended 'Biodiversity Synthesis' group at iDiv helped us to develop the analyses and presentations in several ways. We acknowledge L. Figueiredo and A. Kazem from the iDiv Data & Code Unit for assistance with curation and archiving of the metacommunities resurveys dataset.

W.B.X., S.A.B., J.M.C., R.v.K., and D.M. were supported by the German Centre for Integrative Biodiversity Research (iDiv) Halle-Jena-Leipzig, funded by the German Research Foundation (DFG FZT-118, 20254881); B.J.M. was supported by NSF EPSCOR Track II 201947 and USDA Hatch grant (MAFES #1011538); N.J.G was supported by NSF EPSCOR Track II 201947; I.S.M. was supported by the European Union's Horizon 2020 research and innovation program under Marie Sklodowska -Curie grant (no. 894644); H.S. was supported by Japan Society for the Promotion of Science (JP19K21569 and JP21H03402); A.E.M. and M.D. were supported by Leverhulme Trust (RPG-2019-402). A.F.E acknowledges the Fisheries Society of the British Isles Studentship. BioTIME used here was supported by ERC AdG 250189 and PoC 727440 (to A.E.M.), and Leverhulme Trust Research Centre–the Leverhulme Centre for Anthropocene Biodiversity (to M.D.). Some of the data analyzed here were collected using NSF grants to the LTER Network (NSF06-20443, 8811906, 9411976, 0080529, 0217774, DEB-0423704, DEB-1633026, DEB-1637685, DEB-1256696, DEB-0832652, DEB-0936498, DEB-1832016, DEB-0620652 DEB-1234162, OCE-9982133, OCE-0620959, OCE-1237140, and OCE-1832178).

## Author contributions

W.B.X., J.M.C., S.A.B., D.M., A.E.M., N.J.G., B.J.M., and M.D. conceived the idea for the study; W.B.X., S.A.B., V.B., C.F.Y.C., A.F.E., I.S.M. D.M., F.M., A.S., H.S., R.v.K., A.E.M., N.J.G., B.J.M., M.D., and J.M.C. contributed to data collection; W.B.X. analyzed the data with help from S.A.B.; W.B.X. and J.M.C. wrote the first version of the manuscript, and all authors substantially edited the text.

## Funding

## Competing interests

The authors declare no competing interests.
