## [Peer Review File · Nature Communications]

REVIEWER COMMENTS

Reviewer #1 (Remarks to the Author):

This is an interesting, nicely written, and carefully prepared manuscript. As the authors noted, there is still some uncertainty about the drivers of temporal changes in species occupancy. Most previous studies were based on relatively small spatial and temporal scales, and results are mixed - some show a positive relationship with species range size (widespread species increase their occupancy through time), while others show a negative or lack of relationship. Thus, a synthesis based on the best data available is needed.

The study is based on one major statistical model: 1) a general relationship between range size (the predictor variable) and temporal occupancy change (the response variable). To make things more interesting, the authors added two predictor variables in their analysis to interact with range size: 2) realms (terrestrial, marine and freshwater), and 3) proportion of sites in protected areas.

Results are straightforward to understand: 1) widespread species increase their occupancy through time; 2) this positive relationship is stronger in the marine realm and absent in freshwaters; 3) this positive relationship becomes weaker as the proportion of sites with some protection status increases, but only in the terrestrial realm.

While the idea being pursued here might look simple (and simple is good) at first sight, such kind of synthesis (at this spatial and temporal extent) can be affected by various sources of heterogeneity in the original studies the authors used - 204 time series of metacommunities of multiple taxa spread across the globe monitored for a decade or more. Studies differ in sampling effort, spatial coverage, spatio-temporal replication completeness, etc. And here is the major positive aspect of the study in my opinion: the authors have used a series of different sensitivity analyses to show that the patterns they found hold independent of how they analyze the data. I was positively impressed with the careful selection of data and how the authors dealt with potential sources of heterogeneity in the data that could bias the results. Also, the methods section clearly explains what and how the authors analyzed the data. I am convinced that what they show are general patterns one would usually find if they repeat the analysis. Of course that this does not mean that a small-scale study would not find a negative relationship, for example. But this is what general, large-scale syntheses are: they show general patterns with some level of heterogeneity.

The second positive aspect of the study that I want to highlight is the use of independent data sources to estimate the response and predictor variables. Had they used the same 204 time series to

estimate range size also, the relationships would likely be strongly biased due to lack of independence.

The only major negative aspect of the study is one that the authors also recognize. The analyses do not allow any strong inference about mechanisms to be made. This made the discussion of results a bit speculative and general - maybe even superficial sometimes. The mechanisms driving the patterns they show are likely driving both range size (and there is a whole research field investigating this) and temporal changes in occupancy. At this spatial and temporal scale, only complex, process-based simulations would be able to help us understand such direct and indirect relationships and the role of mechanisms driving them.

So, overall, I think this is a nice piece of work that convincingly shows a generally positive relationship between species range size and temporal changes in species occupancy, which is stronger in the marine realm and tends to be weaker in regions with lots of protected land. Well done!

Having said that, I only have some minor suggestions, comments and doubts.

Title: Do we need this “rich get richer” here? I would drop it. Richness is a term/metric usually associated with communities and here you are referring to species. It would not change the meaning of the title and would make it shorter. Also “widespread species increase [what?] while narrow-ranged species decrease [what?]...”

L. 66. I would write “between range size and occupancy change (or temporal changes in species occupancy)” as you did in L. 69 so that it is clear that temporal changes refer only to occupancy. 
L. 97. 10-km grid cells. That’s big! Do you have metacommunities that are smaller than that? I assume this is the best information available and I know you used an independent source to estimate range size and temporal changes in occupancy, but I wonder if you are missing information (especially) on the narrow-ranged species.

L. 122. Interesting. There is an overall positive relationship, but only in 42% of the studies, the relationship holds. While you suggest that this suggests that different metacommunities conform to each of the hypotheses illustrated in Fig. 1 to varying degrees, it might also be because other variables than range size could be involved with occupancy changes. I think this needs to be discussed.

L. 159. Sorry, I don't get this. Why would the low number of data sets cause a negative relationship within studies?

L. 189. "... species and/or the decrease in occupancy of small-ranged species." In terrestrial ecosystems, right?

L. 223. I was expecting some follow-up on the dispersal issue here.

L. 231. "Our results show that it is indeed that smaller-ranged species which are decreasing in occupancy tend to be replaced...". I know space is limited here, but I miss a discussion about the potential drivers of range size. If the authors have access to traits that are associated with range size, then a more specific analysis considering the traits could help understand the mechanisms. If this is beyond the aims of this study, maybe at least discuss this issue.

L. 260. Not clear to me. So, if the same set of sites included both insects and fish, they were treated as two studies? If so, then you included this dependence in your models?

L. 357. Impressive dataset! Carefully cleaned and organized.

L. 369. "For species with fewer than 3 records...". I am not an expert on species distribution modelling, but are there studies that support the use of this strategy?

L. 579. "'Metacommunity Resurvey' data can be accessed through the iDiv Biodiversity Portal:...". I was not able to access the data through this link.

Tadeu Siqueira

Reviewer #2 (Remarks to the Author):

Xu et al. used globally replicated metacommunity time series datasets to test an interesting hypothesis that long-term changes in species occupancy are explained by species range size. Knowledge of large-scale, long-term changes in ecological communities is still limited. Thus such hypothesis testing with an extensive dataset could provide valuable insight that has the potential to fill this critical knowledge gap. The manuscript is written clearly and also presented in detail. However, due to the inadequacies of the data analysis of this study, I was unable to agree on the validity of the authors' conclusions. For more solid hypothesis testing, the authors need to define occupancy more carefully and adopt strategies of data compiling and statistical analysis that more appropriately account for uncertainty.

General points

- Occupancy must be defined adequately and clearly. The authors expressed occupancy succinctly in Line 63 as the "proportion of sites where a species is present" and gave a more operational definition in Lines 323-325. It is critical that occupancy is a quantity that can be interpreted even when multiple years are included in the early and/or late periods for which occupancy is defined and that there exists a concrete method of calculation that corresponds precisely to the given concept of occupancy. However, I think this requirement for occupancy was not fully met, or at least an adequate explanation for occupancy was not given. The definition of occupancy should be formally expressed in the statistical model to be fitted (Lines 447-448). Still, even there, it was not given enough explanation to allow the reader to determine the occupancy quantity adequately. Because of these deficiencies, I could not determine precisely the change in occupancy that the authors are focusing on and, therefore, cannot support the validity of their conclusions.

- Uncertainty needs to be better accounted for in the analysis of occupancy changes and estimating the effects of protected areas. The authors fitted a hierarchical binomial regression model to late-period data to analyze occupancy changes. In this analysis, the late-period occupancy is modeled as an estimated parameter, but the value of the early-period occupancy is given. Thus, the uncertainty in the changes in occupancy should come from both early period and late period occupancy uncertainty, but the former is wholly ignored in the authors' approach. This could make estimates of the effect on changes in occupancy overconfident. In addition, uncertainty is not adequately addressed in the data compilation for the proportion of protected areas and in estimating the effect of protected areas for which there should be room for improvement (see the following section for details). These issues also raise concerns about the validity of the authors' conclusions.

Specific points

- Lines 138-139. The bracket is not closed.

- Lines 189-191. I didn't understand how this was relevant. The authors seem to expect the slope to be smaller when there are more endemic species and fewer invasive species, but why? Although endemic species might be expected to tend to decrease occupancy and invasive species to increase occupancy over time, the pattern of slope seems non-trivial because species range size should vary in both groups.

- Lines 198-203. I didn't understand the logic of this claim. If the authors think that the power of the significance test was low in freshwater and marine because the data points are biased to the left (Figure 3), I would argue with that. Because the effect of protection is assessed by regressing the effect of range size on the proportion of protected sites, and both freshwater and marine studies are actually sampled in the whole range of 0 to 1, the effect of protection would have been an opportunity to be detected if it existed. The result may instead suggest that the effect of protection differed between realms?

- Line 235. Why is ecological restoration mentioned here? While I do not dispute that ecological restoration is an essential effort in the Anthropocene, this is a factor not addressed in the analysis. Therefore the study cannot logically lead to the claim that ecological restoration is important.

- Lines 265-268 and Supplementary Fig. 6. Although each population is classified into five categories according to patterns of occupancy change, the classification criteria were never explained.

- Lines 323-325. This definition of occupancy is unclear. Do the timescales in the numerator and denominator match? The numerator is a quantity for "a given observation period," and the denominator is a quantity for "each year." A period should often include multiple years. One interpretation I came up with was that the authors calculated occupancy by the proportion of sites where the species appeared at one or more time points in the given period out of the sites surveyed at all time points in the period. However, this contradicts the authors' explanation that they "calculated mean occupancy" for each period (lines 327-328). The above calculation could, at best, give a (potential) upper bound on the annual occupancy that can be observed and does not give a time-averaged quantity. To obtain the mean occupancy for a given period, occupancy must be defined as a quantity associated with the time points within a period. In other words, occupancy must be estimated for each time point, and then its time average must be obtained. In any case, the authors need to state the occupancy definition clearly. The time scale of occupancy must be clarified. The definition of occupancy is critical in this study.

- Lines 413-416. Estimating the degree of protection of a region by the proportion of sites that fall within protected areas is reasonable when a large number of sites are randomly distributed within the area, but will suffer from bias and potentially significant errors when sites are located non-randomly or are limited in number. A more accurate representation should probably be the percentage of the spatial extent of a metacommunity occupied by protected areas. If I understand correctly, this should be immediately calculable from the authors' data set.

- Lines 447-448. In this model, the binomial distribution accounts for the uncertainty of the late-period occupancy. Still, the given value of the early period occupancy $p_{\{1ij\}}$ is incorporated into the offset term without any uncertainty in this quantity being taken into account. This is not a reasonable way to adequately estimate occupancy changes, the study's target variable. Ignoring the uncertainty in $p_{\{1ij\}}$ is likely to underestimate the uncertainty in the change in occupancy $\text{logit}(p_{\{2ij\}}) - \text{logit}(p_{\{1ij\}})$, making it easier for the effect of range to be unduly detected. One possible solution would be to apply a model that specifies a similar binomial distribution for early periods in addition to late periods. This may not be a model class that the brms package can handle, but it should be applied using a generic modeling program such as Stan.

- Lines 449-450. More clarification on nocc and nsamp is required. What quantities are "number of occurrences" and "number of samples?" If observations are made at several time points at a same site, are these data treated separately (in other words, multiple occurrences and samples are offered from the same site)? Or are observations from the same site be pooled into one data point (in other words, one site provides only one occurrence and one sample)? This is closely related to the definition of occupancy and is a critical part of this study.

- Line 453. Different symbols should be used for $\beta_{\{0i\}}$ and $\beta_{\{1i\}}$ to distinguish them from β_0 and β_1 .

- Lines 508-510. The effects of the protected areas were tested by reanalyzing the model fitting results in a meta-analytic model. Such an approach of applying the two models sequentially is not preferred, given that the effect of protected areas on the slope could be estimated by including interaction terms in the hierarchical regression model. If there is no particular reason why a meta-analytic model should be required, then inference based on a single statistical model would provide a more consistent approach.

Reviewer #3 (Remarks to the Author):

Using data from GBIF and a large data set of time series, the authors analysed the relationship between a species' range size and changes in its occupancy through time. This is an important endeavour with implications for ecology, in general, and conservation biology, in particular.

I had two main issues with this paper: (i) One is the conceptual differences between "species' range" and "occupancy". (ii) The other is the statistical model.

(i) if I understood correctly:

- Species' ranges were estimated from GBIF, they correspond to a measure obtained at large spatial scales, and they are assumed to be static over time. Of course, species' ranges change over time, and the authors acknowledge this, but ONLY in lines 373-375 (in the "Methods" section). I think this should be stated much earlier, otherwise, until we get to this point there is a question lurking in our minds: doesn't the species' range size also change in time? I do not disagree with the authors that species' range changes calculated using GBIF for the (median) time range of the time series are likely to be a reasonable first approximation – but this needs to be said early.

- "Occupancy" is a measure obtained from a different data set from that of the species' range (though the authors acknowledge that the same data could have been used in the two data sets – paragraph starting on line 127). "Occupancy" is measured at smaller spatial scales than those of the species' range. Contrarily to the species' range, for the temporal ranges of the time series analysed, occupancy is likely to change.

I think these different temporal and spatial scales implicitly associated with the concept of "species' range" and "occupancy" should be made explicit in the paper as early as possible; I suggest in the Introduction.

(ii) I'm not sure I understood the statistical analyses (but I'm not a statistician). I think it is important to have a better explanation of the model used, and why it was adopted. My main problem is the following:

- What are the implications of using $\text{logit}(p_1)$ as an “offset”? If I understood correctly, by doing so you only “assign” uncertainty to p_2 (modelled with a binomial distribution). Wouldn't it be more appropriate to model directly the difference $(p_2 - p_1)$, which is the quantity you are interested in? (For instance, one could define $x = p_2 - p_1$, and then after a transformation of variables, e.g. $y = (x+1)/2$, and model (y) with a beta distribution.)

Also:

- On lines 433-434 it is said that “The residuals from such a model structure represent changes in occupancy”. But were these residuals ever used in the analyses? (see next comment)

- On lines 447-448 the model is presented, but then on lines 503-504 it seems that you used a simplified version of it to obtain the slopes. The calculation of the slopes is one of the most important parts of the paper, and I'm not sure I fully understand the procedure (e.g. what model was used).

- It certainly makes a lot of sense to use a hierarchical model. However, why use a fixed and a random effect for the same variable? Why didn't you simply use (assuming your approach for p_1 is correct):

$$\text{logit}(p_{2ij}) = \text{offset}(\text{logit}(p_{1ij})) + \beta_{0i} + \beta_{1i} * \text{range}_j$$

- Finally, in the model description, line 448 the variable “range” is indexed with “i” and “j”, but if I understood correctly (and that is why I gave my own interpretation of “species' range” above) range is not a function of the study “i” but only of the species “j”.

To be clear, I'm not saying that your model is wrong but I would like to have a better explanation/motivation for the model.

A smaller detail: How differences in occupancy were calculated is not well explained in the main text, in particular, that it is done using only two points. We find the explanation only in the section “Calculating occupancy”, line 322.

General Response

In the revision, we used a new model with the temporal change in occupancy between the later and early periods as the response variable, incorporating suggestions on modeling from both Reviewers 2 and 3. Our new analyses also included an additional 34 metacommunity time series that were recently added to our database 'Metacommunity Resurveys' (specifically compiled for this and related studies). In addition, we removed two sensitivity analyses from our revision that were no longer necessary. Because our new method did not require the values of 0 or 1 of initial occupancy to be replaced by 0.01 and 0.99, which was necessary for our previous analyses, we removed a sensitivity analysis that used only species with initial occupancy less than 1 and more than 0. We also removed a previous analysis using only assemblage data in the first and last years because resampling two time points from a long-term time series may not provide accurate estimates of biodiversity changes (Stuble et al. 2020, <https://doi.org/10.1002/ecm.1435>). We also made other updates or corrections in response to other comments, and we detail our responses to all comments below. Despite many updates, we found qualitatively consistent results, confirming the conclusion that we reported in the previous version.

Reviewer #1 (Remarks to the Author):

This is an interesting, nicely written, and carefully prepared manuscript. As the authors noted, there is still some uncertainty about the drivers of temporal changes in species occupancy. Most previous studies were based on relatively small spatial and temporal scales, and results are mixed - some show a positive relationship with species range size (widespread species increase their occupancy through time), while others show a negative or lack of relationship. Thus, a synthesis based on the best data available is needed.

The study is based on one major statistical model: 1) a general relationship between range size (the predictor variable) and temporal occupancy change (the response variable). To make things more interesting, the authors added two predictor variables in their analysis to interact with range size: 2) realms (terrestrial, marine and freshwater), and 3) proportion of sites in protected areas.

Results are straightforward to understand: 1) widespread species increase their occupancy through time; 2) this positive relationship is stronger in the marine realm and absent in freshwaters; 3) this positive relationship becomes weaker as the proportion of sites with some protection status increases, but only in the terrestrial realm.

While the idea being pursued here might look simple (and simple is good) at first sight, such kind of synthesis (at this spatial and temporal extent) can be affected by various sources of heterogeneity in the original studies the authors used - 204 time series of metacommunities of multiple taxa spread across the globe monitored for a decade or more. Studies differ in sampling effort, spatial coverage, spatio-temporal replication completeness, etc. And here is the major positive aspect of the study in my opinion: the authors have used a series of different sensitivity analyses to show that the patterns they found hold

independent of how they analyze the data. I was positively impressed with the careful selection of data and how the authors dealt with potential sources of heterogeneity in the data that could bias the results. Also, the methods section clearly explains what and how the authors analyzed the data. I am convinced that what they show are general patterns one would usually find if they repeat the analysis. Of course that this does not mean that a small-scale study would not find a negative relationship, for example. But this is what general, large-scale syntheses are: they show general patterns with some level of heterogeneity.

The second positive aspect of the study that I want to highlight is the use of independent data sources to estimate the response and predictor variables. Had they used the same 204 time series to estimate range size also, the relationships would likely be strongly biased due to lack of independence.

The only major negative aspect of the study is one that the authors also recognize. The analyses do not allow any strong inference about mechanisms to be made. This made the discussion of results a bit speculative and general - maybe even superficial sometimes. The mechanisms driving the patterns they show are likely driving both range size (and there is a whole research field investigating this) and temporal changes in occupancy. At this spatial and temporal scale, only complex, process-based simulations would be able to help us understand such direct and indirect relationships and the role of mechanisms driving them.

So, overall, I think this is a nice piece of work that convincingly shows a generally positive relationship between species range size and temporal changes in species occupancy, which is stronger in the marine realm and tends to be weaker in regions with lots of protected land. Well done!

Reply: Thank you for these positive comments and suggestions. The reviewer is correct that our study cannot provide strong inferences about the mechanisms underlying our findings. This is because, as with any such ‘synthesis’, our results are based on hundreds of heterogeneous datasets from different regions and taxonomic groups. Following specific comments and suggestions below, we have added more discussion on this issue, with specific reference to other possible factors affecting occupancy changes, as well as the possibility to understand mechanisms based on biological traits. Please find responses to specific comments below for details.

Having said that, I only have some minor suggestions, comments and doubts.

Title: Do we need this “rich get richer” here? I would drop it. Richness is a term/metric usually associated with communities and here you are referring to species. It would not change the meaning of the title and would make it shorter. Also "widespread species increase [what?] while narrow-ranged species decrease [what?]...

Reply: Thanks for this suggestion! We have removed “rich get richer” and added “regional occupancy” in the title to make the title shorter and clearer.

L. 66. I would write “between range size and occupancy change (or temporal changes in species occupancy)” as you did in L. 69 so that it is clear that temporal changes refer only to occupancy.

Reply: We have modified the sentence following this suggestion.

L. 97. 10-km grid cells. That's big! Do you have metacommunities that are smaller than that? I assume this is the best information available and I know you used an independent source to estimate range size and temporal changes in occupancy, but I wonder if you are missing information (especially) on the narrow-ranged species.

Reply: In this study, we estimated species' global range size, rather than regional range size or the range size within the extent of metacommunities. While it is true that the metacommunities we surveyed were often smaller than this, our question is about global range size (as a trait of sorts) and its influence on occupancy change. A 10-km (or similar) resolution is often used for estimating global range size in related studies (e.g. Newbold et al. 2018, Staude et al. 2020). Therefore, we feel that the three resolutions (10-, 50-, and 100-km) we used were appropriate. Nevertheless, we acknowledge that this coarse resolution will be limited for very narrow-ranged species. However, only 108 of 16,608 species occupied a single 10-km grid-cell based on GBIF distribution occurrences. In lines 100-103, we have rewritten the sentence as "We estimated species' geographic range sizes as the number of 10-km grid-cells across the world occupied by each species using occurrence records from the Global Biodiversity Information Facility".

L. 122. Interesting. There is an overall positive relationship, but only in 42% of the studies, the relationship holds. While you suggest that this suggests that different metacommunities conform to each of the hypotheses illustrated in Fig. 1 to varying degrees, it might also be because other variables than range size could be involved with occupancy changes. I think this needs to be discussed.

Reply: We agree with this comment. We have added several sentences in lines 140-150 to discuss possible variables or ecological processes that affect occupancy changes: "However, it is also clear that ecological processes other than range size influence changes of species occupancy over time. For example, intrinsic population fluctuations are prevalent in many natural assemblages, which can drive species turnover without external environmental changes²². In addition, biological traits other than range size can influence how species change in their occupancy through time. For example, warm-adapted species are favored relative to cold-adapted species under climate warming²³. Likewise, habitat modification favors non-forest and disturbance-adapted species relative to forest-dwelling and disturbance-intolerant species²⁴. As a result, even though our synthesis based on hundreds of heterogeneous studies shows an overall positive relationship between range size and occupancy change, the large variation around this trend is not unexpected and requires deeper exploration."

L. 159. Sorry, I don't get this. Why would the low number of data sets cause a negative relationship within studies?

Reply: We apologize for the lack of clarity. We intend to express that we have low confidence in the cases with few datasets. There was a negative relationship between range size and occupancy changes in the freshwater system in South America, but only four datasets were sampled. We have clarified this in lines 183-186 in the revision.

L. 189. "... species and/or the decrease in occupancy of small-ranged species." In terrestrial ecosystems, right?

Reply: Yes, you are right. We now clarify that this finding applies to terrestrial ecosystems: "...habitat protection might be successful in stemming some aspects of biodiversity change, in particular, by minimizing the increase in occupancy of large-ranged species and the decrease in occupancy of small-ranged species in terrestrial ecosystems".

L. 223. I was expecting some follow-up on the dispersal issue here.

Reply: In the revision, we have added an example in lines 265-268 to show that large-ranged species are more likely to disperse into regions outside of their native range. As we have no direct evidence and space is limited in the concluding paragraphs, we did not provide more examples or discussion.

L. 231. "Our results show that it is indeed that smaller-ranged species which are decreasing in occupancy tend to be replaced...". I know space is limited here, but I miss a discussion about the potential drivers of range size. If the authors have access to traits that are associated with range size, then a more specific analysis considering the traits could help understand the mechanisms. If this is beyond the aims of this study, maybe at least discuss this issue.

Reply: We agree that clarifying the associations between occupancy changes and traits that are related to range size can help to unravel possible mechanisms. However, this is beyond the aims of our study and appropriate trait data are not available across all the different taxonomic groups in our analyses. In lines 268-283, we have added several sentences to discuss the potential of biological traits. Although some biological traits are related to species range size for specific taxonomic groups, because of the heterogeneity across taxa, general patterns, however, seem not clear in the literature.

L. 260. Not clear to me. So, if the same set of sites included both insects and fish, they were treated as two studies? If so, then you included this dependence in your models?

Reply: This understanding is correct. If both insects and fish were sampled from the same region, they were treated as two studies. There were only seven regions that contained surveys on two or three taxonomic groups, contributing to a total of 15 studies. We now include this information in lines 324-326 in the revision. As we were not specifically examining spatial variation in the relationship between range size and occupancy change, we did not consider any possible non-independence of multiple studies from the same region. However, our model included random effects of the study with varying intercepts and slopes for each study, which allowed for the relationship between range size and occupancy to vary among studies.

L. 357. Impressive dataset! Carefully cleaned and organized.

Reply: Thanks for the positive comment!

L. 369. "For species with fewer than 3 records...". I am not an expert on species distribution modelling, but are there studies that support the use of this strategy?

Reply: Yes, there are studies to estimate ranges using buffered areas around each point for the species with few occurrences, e.g. Guo et al. 2022, *PNAS* (<https://doi.org/10.1073/pnas.2026733119>). It is not possible to construct alpha hulls for species with fewer than three occurrences. We, therefore, used the summed area of 10-km buffers around each point to estimate the extent of occurrences. In lines 469-471, we have rewritten the sentence and added the above literature.

L. 579. "Metacommunity Resurvey' data can be accessed through the iDiv Biodiversity Portal:...". I was not able to access the data through this link.

Reply: We are sorry for this. Because "Metacommunity Resurveys" still has some updates to add (additional datasets and data restructuring), the data have not been uploaded to the iDiv Biodiversity Portal. The final data product will be accessible prior to publication. In the revision, we provide the link (https://github.com/chase-lab/metacommunity_surveys) for the R scripts which were used to compile the "Metacommunity Resurveys". Additionally, all assemblage data used in this study are available on GitHub at https://github.com/Wubing-Xu/Range_size_winners_losers. They will be archived on Zenodo after final acceptance of the article.

Tadeu Siqueira

Reviewer #2 (Remarks to the Author):

Xu et al. used globally replicated metacommunity time series datasets to test an interesting hypothesis that long-term changes in species occupancy are explained by species range size. Knowledge of large-scale, long-term changes in ecological communities is still limited. Thus such hypothesis testing with an extensive dataset could provide valuable insight that has the potential to fill this critical knowledge gap. The manuscript is written clearly and also presented in detail. However, due to the inadequacies of the data analysis of this study, I was unable to agree on the validity of the authors' conclusions. For more solid hypothesis testing, the authors need to define occupancy more carefully and adopt strategies of data compiling and statistical analysis that more appropriately account for uncertainty.

Reply: Many thanks for these constructive suggestions! We have revised the analyses and text throughout the manuscript following these suggestions. Specifically, we clarified the definition of occupancy and occupancy change in both the main text and the Methods section. We updated statistical models to overcome the problems of our previous model. We also added a sensitivity analysis to consider the uncertainty in measuring protection status. Our new analyses produce qualitatively consistent results with those reported before. Please find the details of our responses in addressing each specific comment below.

General points

- Occupancy must be defined adequately and clearly. The authors expressed occupancy succinctly in Line 63 as the "proportion of sites where a species is present" and gave a more operational definition in Lines 323-325. It is critical that occupancy is a quantity that can be interpreted even when multiple years are included in the early and/or late periods for which occupancy is defined and that there exists a concrete method of calculation that corresponds precisely to the given concept of occupancy. However, I think this requirement for occupancy was not fully met, or at least an adequate explanation for occupancy was not given. The definition of occupancy should be formally expressed in the statistical model to be fitted (Lines 447-448). Still, even there, it was not given enough explanation to allow the reader to determine the occupancy quantity adequately. Because of these deficiencies, I could not determine precisely the change in occupancy that the authors are focusing on and, therefore, cannot support the validity of their conclusions.

Reply: We are sorry for the lack of clarity when defining occupancy. We defined occupancy as the proportion of sites where a species is present in a given year. Following this definition, we calculated each species' occupancy within each metacommunity in each year. Because many datasets have data from a few years, we split the time series into two periods and focused on changes in occupancy between the two periods. For the datasets with multiple years within a period, we calculated the occupancy for the period as the average of occupancy from the multiple years. We have clarified the definition of occupancy and the way to calculate it in lines 61-64, 94-100, and 405-431. In the revision, we updated the statistical model, which directly used the occupancy change as the response variable. The occupancy change was the difference in occupancy between the late and early periods. We defined the occupancy change in lines 431-435, and detailed how it was transformed and used in models in lines 542-573.

- Uncertainty needs to be better accounted for in the analysis of occupancy changes and estimating the effects of protected areas. The authors fitted a hierarchical binomial regression model to late-period data to analyze occupancy changes. In this analysis, the late-period occupancy is modeled as an estimated parameter, but the value of the early-period occupancy is given. Thus, the uncertainty in the changes in occupancy should come from both early period and late period occupancy uncertainty, but the former is wholly ignored in the authors' approach. This could make estimates of the effect on changes in occupancy overconfident. In addition, uncertainty is not adequately addressed in the data compilation for the proportion of protected areas and in estimating the effect of protected areas for which there should be room for improvement (see the following section for details). These issues also raise concerns about the validity of the authors' conclusions.

Reply: We agree that our previous model ignored the uncertainty of occupancy in the early period. To address this issue, we updated the statistical model considering a specific suggestion below and a suggestion from reviewer 3. In the new model, we directly used the change in occupancy between late and early periods as the response variable. Because occupancy from both early and late periods was used in the calculation of occupancy change, the new response variable included the uncertainty of occupancy from both early and late periods. Our updated model and description are in lines 542-573 in the revision.

Regarding the uncertainty in measuring the degree to which a metacommunity is protected, we calculated the proportion of the spatial extent of a metacommunity covered by protected areas as another measure following a specific suggestion below. We used the old measure (proportion of sites that fall within protected areas) in the main analyses and the new measure in a sensitivity analysis. To address the concern on the model in estimating the effect of protected areas, we included an interaction between

range size and protection levels in statistical models following another specific suggestion below. These new analyses produced qualitatively consistent results with that shown in the previous version. Please see the responses in addressing the following specific comments for more details. We show these updates in lines 520-530 and 602-622 in the revision.

Specific points

- Lines 138-139. The bracket is not closed.

Reply: We have corrected it. Thanks.

- Lines 189-191. I didn't understand how this was relevant. The authors seem to expect the slope to be smaller when there are more endemic species and fewer invasive species, but why? Although endemic species might be expected to tend to decrease occupancy and invasive species to increase occupancy over time, the pattern of slope seems non-trivial because species range size should vary in both groups.

Reply: We agree that the range size of both endemic and invasive species likely varies within each group. However, endemic species are usually small-ranged, whereas invasive species more often are large-ranged. There are often more threatened and endemic species and fewer invasive species in protected areas, which suggests that protected areas may prevent some threatened and endemic species with small ranges from extirpation and invasive species with large ranges from expanding (clarified in lines 221-224). These observations are consistent with our findings that protected areas may reduce the increase in occupancy of large-ranged species and the decrease in occupancy of small-ranged species in terrestrial ecosystems.

- Lines 198-203. I didn't understand the logic of this claim. If the authors think that the power of the significance test was low in freshwater and marine because the data points are biased to the left (Figure 3), I would argue with that. Because the effect of protection is assessed by regressing the effect of range size on the proportion of protected sites, and both freshwater and marine studies are actually sampled in the whole range of 0 to 1, the effect of protection would have been an opportunity to be detected if it existed. The result may instead suggest that the effect of protection differed between realms?

Reply: Yes, the results suggest that the effect of protection differed among realms. This may be because many marine and freshwater protected areas have not reached their full conservation potential, for example, due to difficulties in enforcing protection and/or emigration of animals outside protection boundaries in highly interconnected habitats. We also think that little or no protection in most regions in marine freshwater realms could reduce our ability to detect any influence of protection in marine and freshwater realms. We have rewritten sentences in lines 235-241 to include this suggestion in the revision.

- Line 235. Why is ecological restoration mentioned here? While I do not dispute that ecological restoration is an essential effort in the Anthropocene, this is a factor not addressed in the analysis. Therefore the study cannot logically lead to the claim that ecological restoration is important.

Reply: We agree with this comment and have removed “ecological restoration” in this sentence.

- Lines 265-268 and Supplementary Fig. 6. Although each population is classified into five categories according to patterns of occupancy change, the classification criteria were never explained.

Reply: We have added the criteria to classify species into five categories in lines 331-338: “By comparing species presence and absence and occupancy changes between late and early periods (see ‘Calculating occupancy’), species were classified into five groups: lost (present in the early period, but absent in the late period), gained (present in the late period, but absent in the early period), persisted with increased occupancy (present in both periods, but higher occupancy in the late period), persisted with decreased occupancy (present in both periods, but higher occupancy in the early period), persisted with stable occupancy (present in both periods with no occupancy changes).”

- Lines 323-325. This definition of occupancy is unclear. Do the timescales in the numerator and denominator match? The numerator is a quantity for "a given observation period," and the denominator is a quantity for "each year." A period should often include multiple years. One interpretation I came up with was that the authors calculated occupancy by the proportion of sites where the species appeared at one or more time points in the given period out of the sites surveyed at all time points in the period. However, this contradicts the authors' explanation that they "calculated mean occupancy" for each period (lines 327-328). The above calculation could, at best, give a (potential) upper bound on the annual occupancy that can be observed and does not give a time-averaged quantity. To obtain the mean occupancy for a given period, occupancy must be defined as a quantity associated with the time points within a period. In other words, occupancy must be estimated for each time point, and then its time average must be obtained. In any case, the authors need to state the occupancy definition clearly. The time scale of occupancy must be clarified. The definition of occupancy is critical in this study.

Reply: We are sorry for the lack of clarity in defining occupancy. The timescales in the numerator and denominator for calculating occupancy were matched. We defined occupancy as the number of sites where a species was present in a given year divided by the total number of sites surveyed in that year. In the last version, we want to use “a given observation period” to refer to the observation period within a year, rather than multiple years. We have clarified the description throughout. In this study, we first calculated the annual occupancy following the above definition. Then we calculated the occupancy for the early and late periods as the average of occupancy from multiple years within that period. Please see lines 405-431 for the definition of occupancy and calculation method in the revision.

- Lines 413-416. Estimating the degree of protection of a region by the proportion of sites that fall within protected areas is reasonable when a large number of sites are randomly distributed within the area, but will suffer from bias and potentially significant errors when sites are located non-randomly or are limited in number. A more accurate representation should probably be the percentage of the spatial extent of a metacommunity occupied by protected areas. If I understand correctly, this should be immediately calculable from the authors' data set.

Reply: Thanks a lot for this suggestion. In this study, we intend to measure the degree to which a metacommunity is protected, and the direct measure is the overall protection status of local sites where assemblage data were collected from. We therefore still used the proportion of sites that fall within

protected areas to represent the degree to which a metacommunity is protected. Inspired by this suggestion, we realized that species occupancy within a given metacommunity is possibly affected by the protection status of the whole spatial extent of a metacommunity because species can disperse across continuous space. Therefore, we also calculated the proportion of the spatial extent of a metacommunity covered by protected areas. We first constructed a convex hull comparing all local sites within a given metacommunity to represent its spatial extent. The convex hull was cropped to keep only areas on the land for terrestrial and freshwater metacommunities and only areas in the sea for marine metacommunities. We then overlaid each cropped convex hull with polygons of protected areas and calculated the proportion of the spatial extent covered by protected areas. In the revision, we used both the proportion of sites that fall within protected areas and the proportion of the spatial extent covered by protected areas. The results were qualitatively consistent (Fig. 3 and Supplementary Fig. 12). In lines 519-529, we have rewritten a paragraph to clarify the metrics to measure the degree to which a metacommunity is protected.

- Lines 447-448. In this model, the binomial distribution accounts for the uncertainty of the late-period occupancy. Still, the given value of the early period occupancy p_{1ij} is incorporated into the offset term without any uncertainty in this quantity being taken into account. This is not a reasonable way to adequately estimate occupancy changes, the study's target variable. Ignoring the uncertainty in p_{1ij} is likely to underestimate the uncertainty in the change in occupancy $\text{logit}(p_{2ij}) - \text{logit}(p_{1ij})$, making it easier for the effect of range to be unduly detected. One possible solution would be to apply a model that specifies a similar binomial distribution for early periods in addition to late periods. This may not be a model class that the brms package can handle, but it should be applied using a generic modeling program such as Stan.

Reply: Thanks a lot for this critical comment. We agree that the uncertainty of occupancy in the early period was not incorporated in our previous model. Considering this comment and one comment from reviewer 3, we now directly calculate occupancy change and use it as the target variable in the revision. The occupancy change was defined as the difference in occupancy between late and early periods. Because most values of occupancy change were distributed around zero, we fit and compared models regressing occupancy change as a function of range size using three error distributions (Gaussian, asymmetric Laplacian and Student' t) in preliminary analyses. However, all these models do not well describe the empirical distribution of occupancy changes. To decrease the kurtosis of occupancy changes, we first square root-transformed the absolute value of occupancy change, and then multiplied that by the sign of the change (termed as 'sign*sqrt-transformation'). The transformed occupancy changes still had values ranging from -1 to 1. We finally model the sign*sqrt-transformed occupancy change as a function of range size using hierarchical linear models. The posterior predictions of the model fitted with Bayesian inference can match the empirical data distribution well. We note that we did not follow the specific model solution that the reviewer suggested here. This is because using occupancy in both early and late periods as response variables in a model will complicate this model and lead to difficulty in interpretation. Instead, our new model uses the occupancy change as the response variable and range size as the explanatory variable, which is matched with the relationship that we aim to assess (the relationship between range size and occupancy change).

Please see lines 485-516 for more details about our new model.

- Lines 449-450. More clarification on nocc and nsamp is required. What quantities are "number of occurrences" and "number of samples?" If observations are made at several time points at a same site, are these data treated separately (in other words, multiple occurrences and samples are offered from the same site)? Or are observations from the same site be pooled into one data point (in other words, one site provides only one occurrence and one sample)? This is closely related to the definition of occupancy and is a critical part of this study.

Reply: Sorry for these unclear definitions. Here, we defined a sample as an observation of a local site within a year. For a site that was observed over several years in a period, it contributed multiple samples in the calculation of occupancy of that period (that is the average of occupancy across different years). We did not pool multiple observations from the same site into one data point. This is because the sites observed in different years were sometimes not in the same locations for some studies, particularly in the marine systems. Although the locations of sites observed in different years may be different, we have matched sites across years to keep them in similar spatial configurations (see '*Data standardization*' in Methods). In the revision, we have clarified the definitions of sample and occupancy in lines 405-431.

- Line 453. Different symbols should be used for β_{0i} and β_{1i} to distinguish them from β_0 and β_1 .

Reply: In the revision (line 567), we have used u_{0i} and u_{1i} instead of β_{0i} and β_{1i} to distinguish them from β_0 and β_1 .

- Lines 508-510. The effects of the protected areas were tested by reanalyzing the model fitting results in a meta-analytic model. Such an approach of applying the two models sequentially is not preferred, given that the effect of protected areas on the slope could be estimated by including interaction terms in the hierarchical regression model. If there is no particular reason why a meta-analytic model should be required, then inference based on a single statistical model would provide a more consistent approach.

Reply: Thanks for this suggestion. We agree that fitting a model including an interaction between range size and the protection level is a better approach. In the revision, we used this approach based on a single model rather than a two-stage model. Nevertheless, the result using the new approach was consistent with the result using the old approach. We found a significant negative interaction between range size and the protection levels in the terrestrial but not in the marine and freshwater realms. Because the main effect of range size was positive, the negative interaction in the terrestrial system means that habitat protection could weaken the relationship between range size and occupancy change. We have revised the text in lines 601-621 and figures (Fig. 3 and Supplementary Figs. 11 and 12) in the revision.

Reviewer #3 (Remarks to the Author):

Using data from GBIF and a large data set of time series, the authors analysed the relationship between a species' range size and changes in its occupancy through time. This is an important endeavour with implications for ecology, in general, and conservation biology, in particular.

I had two main issues with this paper: (i) One is the conceptual differences between “species’ range” and “occupancy”. (ii) The other is the statistical model.

(i) if I understood correctly:

- Species’ ranges were estimated from GBIF, they correspond to a measure obtained at large spatial scales, and they are assumed to be static over time. Of course, species’ ranges change over time, and the authors acknowledge this, but ONLY in lines 373-375 (in the “Methods” section). I think this should be stated much earlier, otherwise, until we get to this point there is a question lurking in our minds: doesn’t the species’ range size also change in time? I do not disagree with the authors that species’ range changes calculated using GBIF for the (median) time range of the time series are likely to be a reasonable first approximation – but this needs to be said early.

- “Occupancy” is a measure obtained from a different data set from that of the species’ range (though the authors acknowledge that the same data could have been used in the two data sets – paragraph starting on line 127). “Occupancy” is measured at smaller spatial scales than those of the species’ range. Contrarily to the species’ range, for the temporal ranges of the time series analysed, occupancy is likely to change.

I think these different temporal and spatial scales implicitly associated with the concept of “species’ range” and “occupancy” should be made explicit in the paper as early as possible; I suggest in the Introduction.

Reply: Thanks for this constructive suggestion. The reviewer is right that different temporal and spatial scales are associated with the species’ geographic range and the occupancy within a metacommunity. In the revision, we have stated this more explicitly in the Introduction (lines 104-109): “For our analyses, we treat species’ geographic range size as a static variable because range expansions or contractions for most species should be very small relative to their global ranges during the relatively short monitoring periods of our study (median = 16 years). By contrast, species occupancy within metacommunities can experience substantial changes over a few decades because it is based on species’ presence and absence at local sites within relatively small regions²¹”

(ii) I’m not sure I understood the statistical analyses (but I’m not a statistician). I think it is important to have a better explanation of the model used, and why it was adopted. My main problem is the following:

- What are the implications of using $\text{logit}(p_1)$ as an “offset”? If I understood correctly, by doing so you only “assign” uncertainty to p_2 (modeled with a binomial distribution). Wouldn’t it be more appropriate to model directly the difference $(p_2 - p_1)$, which is the quantity you are interested in? (For instance, one could define $x = p_2 - p_1$, and then after a transformation of variables, e.g. $y = (x+1)/2$, and model (y) with a beta distribution.)

Reply: We agree that our previous model did not consider the uncertainty of p_1 (occupancy in the early period). Following this suggestion, we calculated the occupancy change $(p_2 - p_1)$ and used a model to focus on explaining the variation of this variable in the revision. We also examined the suggested data transformation $(y = (x+1)/2)$ in a preliminary analysis. Although the transformed variable varied between 0 and 1, its distribution was not well fit by a hierarchical generalized linear model with beta error distribution (i.e. beta regression), because most of its values are distributed around 0.5 (i.e. very high kurtosis). Instead, we first square root-transformed the absolute value of occupancy change, and then multiplied that by the sign of the change (termed as ‘sign*sqrt-transformation’). We model the

sign*sqrt-transformed occupancy change as a function of range size using a hierarchical linear model. The posterior predictions of the model fitted with Bayesian inference can match the empirical data distribution well. Please see lines 542-573 for more details about our new model.

Also:

- On lines 433-434 it is said that “The residuals from such a model structure represent changes in occupancy”. But were these residuals ever used in the analyses? (see next comment)

Reply: In the revision, we have updated statistical models and removed this sentence discussing the residuals. The sentence in the previous version was not clear and we actually did not directly use the residuals in some analyses.

- On lines 447-448 the model is presented, but then on lines 503-504 it seems that you used a simplified version of it to obtain the slopes. The calculation of the slopes is one of the most important parts of the paper, and I’m not sure I fully understand the procedure (e.g. what model was used).

Reply: We apologize for the lack of clarity. The study-level slope estimates were actually obtained from the model shown in lines 447-448 in the previous version. We have updated models (lines 565 - 567) following the suggestions from you and another reviewer in the revision. We still extracted the study-level slope estimates from the model fitting the intercept and slope of range size as both fixed and random effects. In lines 565 -577, we described the new model and specified this is the model where we obtained the study-level slope estimates.

- It certainly makes a lot of sense to use a hierarchical model. However, why use a fixed and a random effect for the same variable? Why didn’t you simply use (assuming your approach for p1 is correct):
$$\text{logit}(p_{2ij}) = \text{offset}(\text{logit}(p_{1ij})) + \beta_{0i} + \beta_{1i} * \text{range}_j$$

Reply: The confusion was probably caused by our lack of clarity in describing the meanings of different coefficients in the previous version. In the formula, we want to highlight global intercept and slope (fixed effects) using β_0 and β_1 and show random effects using u_{0i} and u_{1i} (changed symbols of β_{0i} and β_{1i} following a suggestion of reviewer 2) as the departures of study-level intercepts and slopes from the β_0 and β_1 , respectively. In lines 571-572, we have clarified the meanings of different coefficients.

- Finally, in the model description, line 448 the variable “range” is indexed with “i” and “j”, but if I understood correctly (and that is why I gave my own interpretation of “species’ range” above) range is not a function of the study “i” but only of the species “j”.

Reply: You are right. We have corrected this issue by indexing the variable “range” with only species “j” following this suggestion.

To be clear, I’m not saying that your model is wrong but I would like to have a better explanation/motivation for the model.

Reply: Following your suggestions and the suggestions of reviewer 2, we have used a different model in the revision. In the new model, we directly modeled occupancy change (rather than the occupancy in the late period) as a function of range size. In lines 542-573, you can find the new model and descriptions of this model.

A smaller detail: How differences in occupancy were calculated is not well explained in the main text, in particular, that it is done using only two points. We find the explanation only in the section “Calculating occupancy”, line 322.

Reply: We have added a sentence in the Introduction to define occupancy change (lines 94-100). Occupancy change was defined as the difference in occupancy between the late and early periods. For datasets that only had information from two survey years, the first and last years were used to represent the early and late periods. In the Methods section, we also clarified the definition of occupancy and occupancy change and the approach to calculating them in lines 405-431.

REVIEWERS' COMMENTS

Reviewer #1 (Remarks to the Author):

Thanks for providing responses to my questions and comments. I had already thought that the manuscript was in good shape, and thus now I have nothing else to add. I still think that understanding the drivers of range size should matter here, but if you think this is beyond the scope of your manuscript, then I am good. Well done!

Reviewer #2 (Remarks to the Author):

Thank you for thoroughly addressing the questions and concerns I have raised. I am satisfied with those answers and believe the manuscript is much improved. I was impressed by the meticulous work of the authors, as was the case for the manuscript's initial submission. In this round of peer review, I want to make only a few minor or technical comments. Overall, this work can be considered a novel and excellent study that evaluates the relationship between species occupancy and range size on a global scale.

Page 7, line 136. confidence -> credible?

Page 7, line 138. critical -> credible? Similar errors are found in other parts of the main text and Supplementary Information.

Page 8, line 171. The 90% certainty criterion is applied throughout the manuscript only in this part. Without a rationale for using the 90% intervals, 95% should be used consistently, as in the rest of the manuscript.

Page 21, lines 546-547. How did you confirm that these early models did not adequately describe the observed changes in occupancy? Similarly, how did you check that the models employing the sign * square root-transformation are appropriate? Please describe the procedures for the goodness-of-fit assessment of the models.

Page 22, line 573. standard deviation -> logarithm of standard deviation?

Reviewer #3 (Remarks to the Author):

I congratulate the authors for the efforts made to improve the manuscript, particularly the consideration of a new different statistical model.

At this point I only have a few minor points:

Line 126: Shouldn't it be "Fig. 2"?

Line 136: Given that this study uses Bayesian methods, I suggest using "credible interval" instead of "confidence interval" throughout the paper.

Line 138: What exactly do you mean by "critical interval" (see also line 246)?

Line 567: Although the authors do not explicitly say it, σ_i is in fact being modelled as a power law; maybe this should be stated more explicitly.

Line 608: You refer at this point to something "In the main text", but isn't the Methods section still part of the main text? At least it is not in the Supplementary Material.

REVIEWERS' COMMENTS

Reviewer #1 (Remarks to the Author):

Thanks for providing responses to my questions and comments. I had already thought that the manuscript was in good shape, and thus now I have nothing else to add. I still think that understanding the drivers of range size should matter here, but if you think this is beyond the scope of your manuscript, then I am good. Well done!

Reply: Many thanks for the positive comment.

Reviewer #2 (Remarks to the Author):

Thank you for thoroughly addressing the questions and concerns I have raised. I am satisfied with those answers and believe the manuscript is much improved. I was impressed by the meticulous work of the authors, as was the case for the manuscript's initial submission. In this round of peer review, I want to make only a few minor or technical comments. Overall, this work can be considered a novel and excellent study that evaluates the relationship between species occupancy and range size on a global scale.

Reply: Many thanks for the positive comment.

Page 7, line 136. confidence -> credible?

Reply: We have corrected this.

Page 7, line 138. critical -> credible? Similar errors are found in other parts of the main text and Supplementary Information.

Reply: We have corrected this type of error throughout the main text and Supplementary Information.

Page 8, line 171. The 90% certainty criterion is applied throughout the manuscript only in this part. Without a rationale for using the 90% intervals, 95% should be used consistently, as in the rest of the manuscript.

Reply: We have provided the 95% credible interval in the part (the slope between range size and occupancy change in the terrestrial system) in the revision. Because the terrestrial system had a slope that differed from zero with only 90% certainty, we also provided the 90% credible interval of the slope (line 163).

Page 21, lines 546-547. How did you confirm that these early models did not adequately describe the observed changes in occupancy? Similarly, how did you check that the models employing the sign * square root-transformation are appropriate? Please describe the procedures for the goodness-of-fit assessment of the models.

Reply: We checked the model qualities by comparing kernel density estimates of observed occupancy changes and predicted ones from different models based on posterior predictive checks using “pp_check” function in the R package ‘brms’. We have described the procedures in lines 513-514 and also added the Supplementary Fig. 17 to show the results of the comparison in the revision.

Page 22, line 573. standard deviation -> logarithm of standard deviation?

Reply: We have updated the text following this suggestion.

Reviewer #3 (Remarks to the Author):

I congratulate the authors for the efforts made to improve the manuscript, particularly the consideration of a new different statistical model.

Reply: Many thanks for the positive comment.

At this point I only have a few minor points:

Line 126: Shouldn't it be “Fig. 2”?

Reply: Yes, you are correct. We have corrected this.

Line 136: Given that this study uses Bayesian methods, I suggest using “credible interval” instead of “confidence interval” throughout the paper.

Reply: We have replaced “confidence interval” with “credible interval” throughout the main text and Supplementary Information.

Line 138: What exactly do you mean by “critical interval” (see also line 246)?

Reply: It indicates “credible interval”. Sorry for the typo. We have corrected this.

Line 567: Although the authors do not explicitly say it, σ_i is in fact being modelled as a power law; maybe this should be stated more explicitly.

Reply: You are right. We have stated that we modeled the logarithm of standard deviation of occupancy change as a function of the log-transformed number of samples (lines 529 and 540).

Line 608: You refer at this point to something “In the main text”, but isn't the Methods section still part of the main text? At least it is not in the Supplementary Material.

Reply: Here it indicates Fig. 3. We have specified this in the text.